# Electrochemical Polishing of Ti6Al4V Alloy Assisted by High-Speed Flow of Micro-Abrasive Particles in NaNO_3_ Electrolyte

**DOI:** 10.3390/ma15228148

**Published:** 2022-11-17

**Authors:** Jia Liu, Zhen Wang, Zhengyang Xu

**Affiliations:** College of Mechanical and Electrical Engineering, Nanjing University of Aeronautics & Astronautics, 29 Yu Dao Jie Street, Nanjing 210016, China

**Keywords:** electrochemical polishing, flowing abrasive particles, titanium alloy, passivation film, surface quality

## Abstract

Electrochemical polishing (ECP) is an efficient and low-cost technology for polishing difficult-to-machine materials with complex structures. However, when an environmentally friendly neutral salt solution is used as the polishing electrolyte, a dense passivation film forms on the surface of passive metals, such as titanium alloy, with a serious detrimental effect on the polishing efficiency and surface quality. In this paper, we introduce an ECP method assisted by a high-speed flow of micro-abrasive particles (ECFAP). The contribution of the flowing micro-abrasive particles in the ECP process enables the electrochemical dissolution and abrasive polishing to occur simultaneously on the workpiece surface. The high-speed abrasive particles remove the passivation film formed under ECP, thereby improving the polishing efficiency and quality. We carried out the comparative tests of conventional ECP and the proposed ECFAP on a Ti6Al4V alloy in 10% NaNO_3_ electrolyte; the results show that, while the matrix material forms a soft high-impedance passivation film under ECP, this film is removed by the high-speed flowing abrasive particles under ECFAP. The proposed ECFAP method improves both the polishing efficiency and the surface quality. Finally, ECFAP-treated specimens with an optimum voltage of 3 V for 10 min exhibited an average surface roughness of 0.0953 µm.

## 1. Introduction

Polishing improves the surface quality of machined workpieces, eliminates surface fatigue cracks, reduces friction and wear, significantly improves the working performance and service life of products [1,2,3,4,5,6], and currently plays an important role in advanced manufacturing. Traditional polishing technology mainly involves mechanical burnishing (MB) and grinding [7,8]. However, the continuous increase in industrial production has resulted in the emergence of many difficult-to-machine materials and complex structures, whose polishing represents a challenging task. To improve the efficiency and quality of the polishing process, many nontraditional and composite polishing technologies have been proposed, such as abrasive flow machining (AFM) [9,10,11,12,13], chemical polishing (CP) [14], magnetic abrasive finishing (MAF) [15,16,17], laser polishing (LP) [18,19], and electrochemical polishing (ECP) [20,21]. Table 1 lists the characteristics and applicable environments of these polishing technologies. Polishing technologies for the complex structures of difficult-to-machine materials have always received much attention from both industrial and academic researchers.

ECP is a nontraditional machining technology to improve surface quality by exploiting the different electrochemical dissolution rates at the convex and concave points of a surface. Compared with other polishing technologies, ECP has the advantages of high efficiency and low cost in polishing complex structural parts made up with difficult-to-machine materials. At present, ECP is generally carried out in an acidic solution. Simka et al. [22] analyzed the corrosion resistance of the Ti-13Nb-13Zr alloy and determined that a bath composed of sulfuric acid and ammonium fluoride had the best ECP performance. Wang et al. [23] studied the ECP of a three-dimensional printed titanium alloy with an acid alcohol solution, optimized the process parameters, and obtained a surface roughness (Ra) of 0.3 μm. To explore the use of a neutral salt solution as a more environmentally friendly electrolyte, Xu et al. [24] employed polarization curves and open-circuit potentials to compare the electrochemical dissolution behaviors of a Ti60 titanium alloy in NaNO_3_ and NaCl solutions at different concentrations and temperatures. The results showed that a dense passivation film (PF) formed during the electrolysis of the titanium alloy, hindering the electrochemical reactions. To remove the PF and improve the surface quality, Wang et al. [25] studied the electrochemical dissolution characteristics of TC17 in NaNO_3_ solution. The results showed that the PF was removed in a short time at a high current density, producing a well-machined surface. Hryniewicz et al. [26,27] studied the surface characteristics of 316L stainless steel after ECP in a magnetic field, and analyzed the influence of the magnetic field on the PF. However, high current densities can lead to excessive electrolytical corrosion of the material surface, altering the contour shape and affecting the accuracy of machined parts. Electrochemical–mechanical combined polishing (EMCP) has proved to be an effective solution for removing the PF. Mechanical grinding removes the PF at convex points of the surface; then, electrochemical corrosion removes the exposed material in convex points to perform the polishing. An et al. [28] proposed an EMCP process using a tool assembly, comprising a coarse and a fine grinding wheel along with a cathode. This allowed the polishing of the interior of a curved 18 mm-diameter channel, resulting in smooth surfaces for both the curved channel and a corresponding straight channel. The EMCP process can use a neutral salt solution as an environmentally friendly electrolyte to achieve better polishing; however, the accessibility of the grinding tools to complex structures may be limited, thus affecting the polishing quality. With the widespread use of the additive manufacturing of complex structural parts, the demand for polishing in environments with poor tool accessibility is increasing. A convenient, efficient, and low-cost polishing technology is urgently needed.

To simplify the design and improve the accessibility of the EMCP grinding tools, in this study we propose an ECP method assisted by a high-speed flow of micro-abrasive particles (ECFAP). This method removes the PF through abrasive polishing by high-speed flowing particles, without grinding tools. The combination of electrochemical dissolution and abrasive polishing by flowing particles significantly improves the efficiency and quality of the ECP with a neutral salt electrolyte. The main purpose of this work is to explore the influence of the high-speed flowing abrasive particles on the polishing efficiency and quality. The polishing mechanism of the ECFAP process was studied using polarization curves as well as electrochemical impedance spectroscopy (EIS), micromorphology, energy-dispersive X-ray spectrometry (EDX), and surface roughness measurements. Finally, comparative ECP and ECFAP experiments were carried out to explore the effect of the high-speed flow of abrasive particles on the polishing quality and efficiency.

## 2. Principles of ECP and ECFAP Methods

Figure 1 shows the mechanism of conventional ECP using a neutral salt electrolyte. The workpiece and cathode are connected to the positive and negative pole of the power supply, respectively, and the neutral salt solution used as electrolyte flows at high speed into the polishing gap between the tool and the workpiece. When the two poles of the power supply are energized, a dense oxide film forms on the surface of the workpiece; this passivation layer hinders the electrochemical reaction and has a serious detrimental effect on the polishing efficiency. In addition, the local weak areas of the PF break under the scouring action of the electrolyte and the local dissolution of the material causes pitting corrosion that also has a serious detrimental effect on the polished surface quality.

To solve the problems associated with the PF during conventional ECP, in this study we propose the ECFAP method, whose polishing mechanism is shown in Figure 2. Similar to conventional ECP, a polishing gap is present between the cathode tool and the anode workpiece. Abrasive particles and electrolyte are mixed in a mixing tank to direct a high-speed two-phase flow into the polishing gap using a centrifugal pump; during machining, the high-speed flowing abrasive particles scour the workpiece surface. The particles themselves have a leveling effect on the substrate surface, removing the PF and accelerating the electrochemical dissolution at convex points on the surface. In this way, the convex points of the substrate surface undergo cyclic passivation, depassivation, and electrochemical dissolution, thereby improving the surface quality.

## 3. Experimental Procedures

### 3.1. Electrochemical Tests

Titanium alloys are often used in the manufacturing of key aerospace components but are difficult to machine and exhibit strong passivation during electrochemical machining (ECM). Therefore, the titanium alloy Ti6Al4V (TC4) was selected as substrate to compare the dissolution behaviors of conventional ECP and the proposed ECFAP. Polarization and cyclic voltammetry (CV) curves as well as EIS data of TC4 were obtained using an electrochemical workstation (Zennium E, Zahner, Germany); a typical three-electrode system was used in the experiment. The working electrode was a TC4 cube with dimensions 10 mm × 10 mm × 10 mm, and the TC4 composition is shown in Table 2. During the tests, only one surface was exposed to 10 wt.% NaNO_3_ (25 ± 0.5 °C) and the other was covered with an insulating epoxy resin. A saturated calcium electrode and a platinum plate (20 mm × 20 mm × 0.2 mm) were used as a reference and auxiliary electrode, respectively. The measurements were carried out in the frequency range from 100 kHz to 10 MHz, and the amplitude of the sinusoidal signal was 10 mV.

### 3.2. ECFAP Experiments

Figure 3a shows a schematic diagram of the experimental setup, whose main components were a liquid mixing tank, a stirrer, a power supply, and a fixture. In the liquid mixing tank, a 10 wt.% NaNO_3_ solution and 500-mesh SiC abrasive particles were mixed in a volume ratio of 9:1. To fully mix the electrolyte and abrasive particles, a slurry stirrer constantly operated in the liquid mixing tank during the experiment. A pump injected the mixed solution at a certain pressure into the fixture. In this work, the workpiece and the cathode tool were cubes with a cross-section of 15 mm × 15 mm. They were held face-to-face in the cube recesses of the fixture, and both had an exposed area of 15 mm × 15 mm in the machining region. In the experiment, no relative motion occurred between workpiece and cathode tool and the machining gap (the distance between cathode tool and workpiece was 2 mm).

Because the flow channel structure has a significant effect on the hydrodynamic state of the mixed solution [29], a continuous segmented constricted flow channel was designed inside the fixture, as shown in Figure 3b. The cross-sectional areas of the nozzle, inflow section, and uniform flow sections were 150, 75, and 30 mm^2^, respectively. The parameters of the ECFAP experiments are shown in Table 3.

The tests were carried out on a surface machined by electrical discharge machining (EDM). The influence of the high-speed flow of abrasive particles on the quality and efficiency of the electrolytic finishing was explored by measuring the surface micromorphology and roughness. Then, based on the polarization curve of TC4 under ECM, ECFAP experiments were carried out under appropriate voltages corresponding to the activation, passivation, and over-passivation regions and the mechanism of the ECFAP process was investigated.

### 3.3. Surface Integrity Measurements

A comprehensive analysis of the surface integrity characteristics of machined TC4 alloy samples was then performed, including surface morphology, elemental composition, nanoindentation, 3D profiling, and surface roughness tests. The surface morphology was analyzed by scanning electron microscopy (SEM, S4800, Hitachi, Tokyo, Japan) and secondary electron imaging SEM with an acceleration voltage of 15 kV. Element contents were measured by an EDX spectrometer (X-flash 5010, Bruker, Karlsruhe, Germany) in point-scanning mode. The surface mechanical properties of the samples were measured by a nanoindenter (U9820A, Agilent, Santa Clara, CA, USA). During the measurements, the loading depth was 2 μm and the loading–unloading curve and hardness were recorded. The probe pressing rate during loading was set to 6 nm/s, and the data were recorded every 0.2 s. In addition, 3D profiles and surface roughness were measured with a 3D optical profiler (VHX-6000, Keyence, Tokyo, Japan) and a surface finish tester (M300C, Mahr, Esslingen, Germany), respectively. The sampling length of the surface roughness was set to 5.6 mm, according to the dimensions of the workpiece.

## 4. Results

### 4.1. Polarization and CV Curves

To explore the polarization behavior of TC4, polarization and CV curves were obtained in 10% NaNO_3_ solution, as shown in Figure 4. The measurements were carried out at potentials ranging from −2 to 14 V. Figure 4a shows that both ECM and ECFAP involved a passive and a transpassive region [22,25]. The dissolution of the anode material is hardly observed in the passive region, which is attributed to the formation of a PF on the surface of the matrix material. When the voltage exceeds the decomposition threshold, the PF is destroyed, the matrix material begins to dissolve, and the current rises sharply. The corrosion potential of TC4 was 11.2 V under ECP and dropped to 4.2 V under ECFAP conditions; the reason for this change is that the PF was removed by the high-speed flow of abrasive particles, thereby accelerating the electrochemical dissolution and reducing the dissolution potential.

The CV curve of TC4 is shown in Figure 4b; the measurements were carried out in the potential range from −2 to 15 V. In the forward scanning process, the matrix material began to dissolve from 11.2 V; after the dissolution, the current density showed an almost linear relationship with the voltage. The reverse scan of the CV curve began when the voltage exceeded 15 V, and the current density decreased with decreasing voltage. When the voltage decreased to 4.2 V, the material surface was passivated again, and the current density dropped to zero. The reverse scan curve was positioned above the forward scan one, due to pitting on the surface of the workpiece; the product located in the pits mixed with the electrolyte, resulting in a higher conductivity than that of the solution.

### 4.2. Analysis of Passivation Film

To further explore the characteristics of the PF, the surface structure of the matrix material was studied by EIS. The elemental composition of the PF surface was analyzed by EDX, and the passivation surface was measured by nanoindentation tests. Moreover, the impedance of the PF was measured using an electrical equivalent circuit (EEC). To study the effect of the voltage on the PF, the matrix material was passivated rapidly in 10% NaNO_3_ solution at 4, 6, and 8 V for 10 min. Then, EIS tests were conducted immediately under each condition, and the Zview software package was used to fit the EIS data. Figure 5a,c,d show the EIS curves of TC4 in 10% NaNO_3_ solution; the multiple-peak phase angle plot at 8 V in Figure 5a indicates the existence of at least two time constants. Therefore, the EEC model shown in Figure 5b was used to fit the EIS data. The impedance of the constant phase element (CPE) is defined as:Z_CPE_ = [Q(*i_u_*w)*^n^*]^−1^(1)
where Q is the CPE constant, *w* is the angular frequency (rad/s), *i_u_* is the imaginary unit (*i_u_*^2^ = −1), and *n* is the dispersion index of the CPE. The latter is related to the surface roughness of the blank or to the uneven distribution of the PF, and its values range between 0 and 1. When *n* = 0.5, the CPE corresponds to the Warburg impedance, whereas when *n* = 1, it represents the ideal capacitance; when 0.5 < *n* < 1, the CPE is intermediate between ideal capacitance and Warburg impedance.

The EIS fitting results are shown in Table 3. The fitting curve reproduces well the experimental data points, as confirmed by the small *χ*^2^ value in Table 4. The table also shows the fitting parameters of the EEC model for TC4, where *R_S_* is the solution resistance of 10% NaNO_3_, *R_ct_* is the polarization resistance of the electrode–solution interface, and *R_film_* is the resistance of the PF. The results show that *R_film_* was 2378, 173,800, and 231,190 Ω cm^2^ at a potential of 4, 6, and 8 V, respectively, which indicates that increasing the potential within the passivation range increased the resistance and thickness of the PF.

In addition, EDX was used to analyze the elemental composition on the PF surface. The workpiece was immersed in 10% NaNO_3_ solution and passivated at 6 V for 2 min to form a stable PF, and EDX elemental analysis was carried out on the initial and passivated surfaces. Figure 6a shows that the mass and atomic fractions of oxygen on the surface of the original blank sample were 0.14% and 38%, respectively; after passivation, the corresponding values were 8.08% and 19.47%, respectively. This confirms that the PF consisted of the oxide formed by TC4 during electrolysis. This layer was attached to the workpiece surface and had high resistance, inhibiting the electrochemical dissolution of the substrate. Furthermore, continuous stiffness measurements of the original and ECM-passivated surfaces were performed at 6 and 12 V using a nanoindenter. The measured load–displacement curves and the calculated stiffnesses are shown in Figure 7. According to the load–displacement curves, the load after anodization was significantly lower than that of the original surface with the same displacement. The average hardness of the original TC4 substrate material was 4.2 GPa, and decreased to 3.5 GPa at 6 V and 2.4 GPa at 12 V. Therefore, the average hardness of the TC4 surface with a PF was lower than that of the substrate material. Combining the EIS, EDX, and nanoindentation results, it can be concluded that, during the electrolysis process, TC4 forms a soft PF consisting of an oxide layer attached to the surface of the workpiece, which affects the mechanical properties of the surface and inhibits electrolytic reactions.

To further understand the mechanism of PF formation and dissolution by ECFAP, tests with different ECP and flowing abrasive polishing (FAP) combined pretreatments were analyzed by EIS. The specific pretreatment methods are described in Table 5. According to the parameters and results obtained in the previous tests, the preprocessing time of both ECP and FAP was set to 5 s. Workpieces 1 and 2 were compared to assess whether the high-speed flow of abrasive particles removed the PF, whereas the comparison of workpieces 2 and 3 was used to assess whether the PF was regenerated after being destroyed by the particles; moreover, workpieces 3 and 4 were compared to assess again whether the flowing particles removed the PF.

The EIS results are shown in Figure 8. Table 6 lists the values of the parameters obtained by fitting to the EEC, as shown in Figure 8b. Figure 8c shows that, at a relatively low frequency (0.01 Hz), the phase angle after machining was close to −40°, whereas the phase angle diagram for the ECP under 6 V (Figure 4) shows that the corresponding phase angle was ca. −45°.

The radius of the capacitive arc is an important indicator of the corrosion resistance, with a larger radius denoting a greater resistance to charge conduction and a higher corrosion resistance [30]. The radius of the capacitive arc of the samples after FAP (workpieces 2 and 4) was smaller than that after ECP (workpieces 1 and 3). On the other hand, the *R_film_* values of workpieces 1, 2, 3, and 4 (Table 5) were 113,890, 1983, 116,820, and 3074 Ω cm^2^, respectively. The comparison of the EIS data for workpieces 1 and 2 shows that the corrosion resistance of the samples decreased after both ECP and FAP. This is attributed to the removal of the PF by the abrasive particles; the same conclusion was reached in the studies published by Yang et al. and Chen et al. [31,32]. Similarly, the comparison of the EIS data of workpieces 2 and 3 reveals that, after another ECP treatment, the PF was regenerated on the surface, and the corrosion resistance was improved. In conclusion, the PF underwent continuous formation and removal during ECFAP.

In conclusion, the EIS measurements show that TC4 formed a PF during ECP, which hindered the electrochemical reactions, while the nanoindentation and EDX measurements reveal that the PF consisted of a soft oxide layer. Moreover, the EIS measurements show that the high-speed flow of abrasive particles removed the PF and reduced the impedance of the substrate in ECP.

### 4.3. Results of ECFAP Experiments

To demonstrate the feasibility of the proposed method, we carried out ECP and ECFAP experiments on a sample surface obtained by EDM. Figure 9a shows that the original EDM surface exhibited recrystallized species and craters. After ECFAP at 12 V for 60 s, the recrystallized species and craters disappeared, as shown in Figure 9b; the surface quality of the workpiece improved, with evident grain boundaries, and the Ra value decreased from 4.410 to 0.945 μm. Under the same parameters used for the ECP, the surface quality was not improved and limited electrochemical dissolution was observed, as shown in Figure 9c. When the voltage was increased to 35 V, serious pitting corrosion occurred on the surface, whose roughness increased significantly, as shown in Figure 9d; this was because the PF inhibited electrochemical reactions and uneven corrosion appeared on the workpiece surface even at higher voltages.

Figure 10 shows that the current density during the ECFAP treatment at 6, 9, and 12 V was more than 10 times higher than that during ECP. Combined with the processing effect and the comparison of the current values measured during different processes, these results show that the high-speed flow of abrasive particles evenly removed the PF, improving the current density and the polishing quality.

We then explored the influence of the flowing abrasive particles on the efficiency of the ECP process. The time evolution of the surface roughness is shown in Figure 11; at a voltage of 6 V, the Ra value of the ECP in 10% NaNO_3_ solution remained constant at ca. 0.85 μm. The occurrence of micro-pitting corrosion on the material surface prevented the improvement of the surface quality. Under FAP conditions, the surface roughness decreased slowly to approximately 0.7 μm after 8 min, whereas the surface roughness of the ECFAP-treated sample showed a rapid drop to less than 0.5 μm within 1.5 min. As the ECFAP treatment continued, electrolytic corrosion gradually became dominant, and the surface quality deteriorated rapidly due to micro-pitting corrosion. Therefore, the processing time significantly influences the surface quality achieved with the ECFAP method.

The dissolution mechanism of the high-speed flow of abrasive particles in the activation, passivation, and over-passivation regions was explored based on the polarization curves. Figure 12 shows the material removal mechanism of the ECFAP method in the activation region under 3 V. The initial surface contained several scratches; after ECFAP for 2 min, the scratches were significantly reduced and the quality of the machined surface was improved. After 5 min, most of the workpiece surface was flat, with only a small amount of residual scratches. After 10 min, the scratches had disappeared and the surface quality was good. No pitting corrosion of the matrix could be observed at 500× magnification.

The comparison of the efficiencies of FAP and ECFAP under different voltages in the activation region [Figure 13a] revealed that ECFAP achieved a significantly faster reduction of the surface roughness. In the activation region, a greater voltage corresponds to a higher polishing efficiency and better surface quality. By monitoring the current during ECFAP at 3 V [Figure 13b], we found that the current density was ca. 50 A/dm^2^; although this value was small, it still indicated the occurrence of a small amount of dissolution.

In the activation region, the high-speed abrasive particles ground the substrate surface and improved its quality, and the matrix continuously generated a soft PF in 10% NaNO_3_ solution; the abrasive particles also removed the PF. After the latter process, a small amount of electrochemical dissolution occurred at surface protrusions, improving the polishing efficiency of the workpiece surface. After the PF removal by the high-speed abrasive particles, a new PF was immediately formed and prevented excessive dissolution of the matrix material. Therefore, the final machined surface had no pitting corrosion and good surface quality.

Figure 14 shows the results of the ECFAP experiment in the passivation region under 6 V. The original morphology of the workpiece surface presented many abrasive polishing marks and the scratches could still be seen under 500× magnification. After ECFAP for 30 s, the scratches were obviously reduced and the electrochemical dissolution marks could be barely seen. When the machining time reached 60 s, only a few wear marks were visible and the electrochemical dissolution marks became increasingly obvious. When the ECFAP time reached 90 s, the wear marks disappeared and the electrolytic surface replaced the original surface. Electrochemical dissolution marks could be seen on the surface at 500× magnification. After ECFAP, the surface had a flat morphology and the metallographic structure of the material was barely visible. In the passivation region, the high-speed flow of abrasive particles removed the PF. On one hand, the particles polished the substrate surface; on the other hand, they removed the PF and accelerated the uniform dissolution of the matrix material. The morphology of the dissolved material contained micro-pits. Finally, the electrolytic surface replaced the original scratched surface, improving its quality.

In the over-passivation region, ECP and ECFAP experiments were carried out at voltages under 35 V. The surface microstructure after machining is shown in Figure 15. Under 35 V, the ECP surface mostly consisted of pits and its quality was poor, whereas the ECFAP-treated surface did not exhibit a pitted morphology and its dissolution was uniform. The surface quality improved after ECFAP; the reason for this improvement is that the potential penetrated through the PF and the dissolution spread from breaches in the PF to their surroundings. The potential could not penetrate the whole PF, and the latter was continuously regenerated during the polishing process. Therefore, the surface treated by ECP under 35 V was uneven and its quality was poor. In contrast, upon ECFAP treatment, the high-speed flow of abrasive particles evenly removed the PF and dissolved the matrix, resulting in a good surface quality after polishing.

Based on the results of the ECFAP experiments, it can be concluded that the high-speed flow of abrasive particles was able to uniformly remove the PF and accelerate the dissolution of the substrate material, a phenomenon that could be observed at higher current densities. The processing quality and efficiency were significantly improved by the high-speed flow of abrasive particles. The initial surface was polished to a roughness of Ra = 0.3 μm using abrasive paper. By optimizing the parameters, an ECFAP test was performed at 3 V for 10 min. Figure 16 shows the 3D morphology and roughness profile of the sample surface before and after ECFAP. The surface roughness of each specimen was measured along four measuring lines in two horizontal and two vertical directions. The cutoff length and linear scanning length were 0.8 and 5.6 mm, respectively. The average roughness decreased from 0.2988 to 0.0953 μm after ECFAP, with mean square deviations of 0.0223 and 0.0101 μm, respectively. After ECFAP treatment, the phase structure of the machined surface was detected by X-ray diffractometer (PANalytical X’pert, PANalytical, NL), as shown in Figure 17. It should be noted that there is little difference between ECFAP treatment and grinding surface. It can be inferred that there is no obvious residual electrolytic layer on the sample surface after ECFAP treatment.

## 5. Discussion

A model illustrating the removal of TC4 matrix material by ECFAP is shown in Figure 18. The main principle of the ECFAP process is that the matrix material forms a PF upon electrolysis in 10% NaNO_3_ solution. Under the action of the high-speed flow of abrasive particles, the PF on protrusions of the workpiece surface will be removed, and the electrochemical anodic dissolution reaction will thus be focused on those protrusions. In contrast, the PF on valley regions will be retained and protect the matrix material from corrosion. Therefore, the protrusion sites will dissolve first, flattening the surface of the workpiece.

The material removal models by ECFAP in different voltage ranges corresponding to activation, passivation, and over-passivation are illustrated in Figure 19. In the activation region, the high-speed flow of abrasive particles removes the soft PF and micro-protrusions on the surface of the substrate material. This accelerates the dissolution rate at the protrusions. The polished surface exhibits no pitting corrosion and an improved surface quality. Therefore, this voltage range is suitable for polishing. The surface evolution with polishing time under different voltages has been discussed above. In short, at a small voltage, as the polishing time increases, the surface tends to be smooth and gradually tends to be stable. As shown in Figure 16, a better surface quality is obtained at a voltage of 3V and a processing time of 10 min. In the passivation region, the workpiece surface is always in a passivated or trans-passivated state. When the PF is removed by the flowing abrasive particles, the protrusions are first dissolved into flat regions and then corroded into pitting areas, which prevents the improvement of the surface roughness. Within this voltage range, a better surface quality can be obtained by adjusting the polishing time. With the extension of polishing time, the polishing effect decreased rapidly and then increased rapidly, indicating that short-time mixing polishing effect is better. In the over-passivation region, the removal of the PF by the flowing abrasive particles still accelerates the dissolution of the substrate material. However, higher dissolution rates can be obtained due to the higher voltage established between the anode and cathode. The higher operating voltage established in the electrochemical system enhances pitting and increases the surface roughness. Therefore, this voltage range is only suitable for the rough polishing of surfaces with poor surface quality.

## 6. Conclusions

This paper presents an ECP method assisted by a high-speed flow of abrasive particles. The combination of electrochemical dissolution and abrasive polishing by the flowing particles significantly improved the efficiency and quality of ECP using a neutral salt electrolyte.

Polarization and CV measurements showed that the TC4 surface was passivated in the ECP process. In the passivation region, the matrix material hardly dissolved: nanoindentation and EDX analyses revealed that a soft PF was formed on the surface of the workpiece and hindered electrochemical reactions. Combined with EIS measurements, the results indicated the presence of a high-impedance PF, and a structural model of the TC4 PF and matrix material was obtained. In addition, based on the values of the EIS parameters, it was shown that the high-speed flow of abrasive particles under ECFAP removed the PF and reduced the impedance.

The polishing effect of ECFAP at voltages within the activation, passivation, and over-passivation regions was investigated based on the polarization curves of TC4 under ECP. The tests showed that the high-speed flow of abrasive particles accelerated the dissolution of the substrate material in all voltage regions. In the passivation and over-passivation regions, pitting corrosion occurred on the substrate surface as the dissolution rate and voltage increased, preventing the improvement of the surface roughness. At voltages within the activation region, no pitting corrosion occurred on the ECFAP surface and the best polishing results were achieved. Finally, ECFAP-treated specimens at a voltage of 3 V for 10 min exhibited an average surface roughness of 0.0953 µm.

## Figures and Tables

**Figure 1 materials-15-08148-f001:**
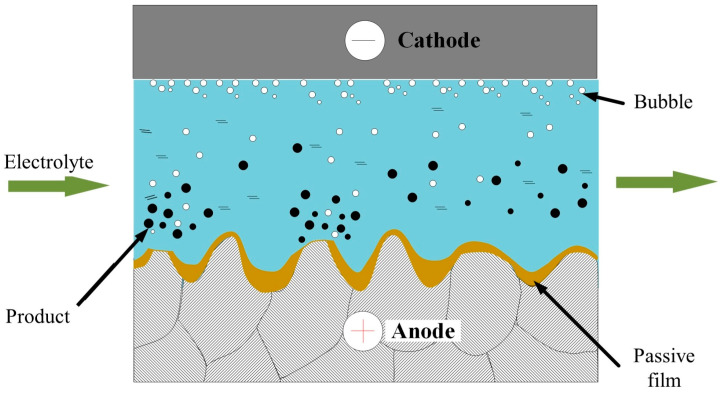
Polishing mechanism of conventional electrochemical polishing (ECP).

**Figure 2 materials-15-08148-f002:**
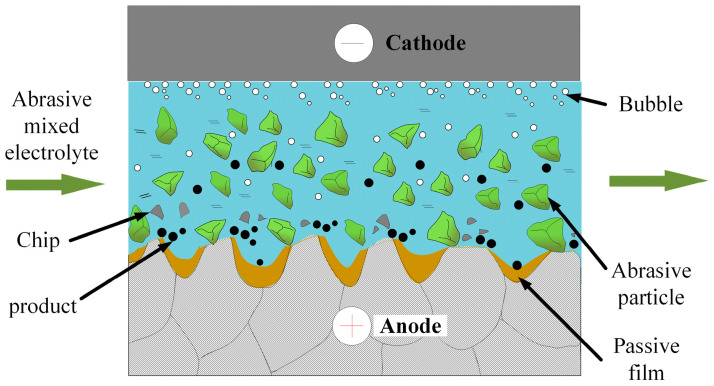
Polishing mechanism of proposed ECP method assisted by high-speed flow of micro-abrasive particles (ECFAP).

**Figure 3 materials-15-08148-f003:**
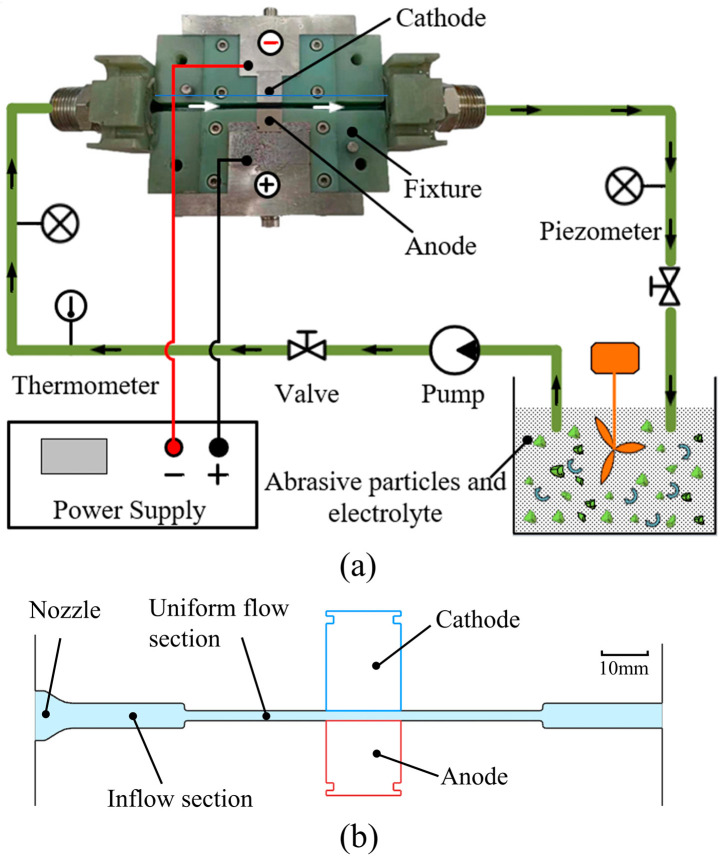
Schematic illustration of experimental setup. (**a**) experimental system; (**b**) internal flow channel structure of fixture.

**Figure 4 materials-15-08148-f004:**
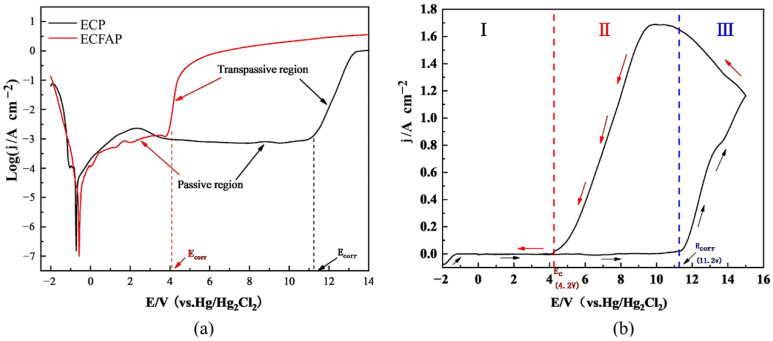
Polarization and cyclic voltammetry (CV) curves of TC4. (**a**) polarization curves under ECP and ECFAP conditions; (**b**) CV curve under ECP conditions.

**Figure 5 materials-15-08148-f005:**
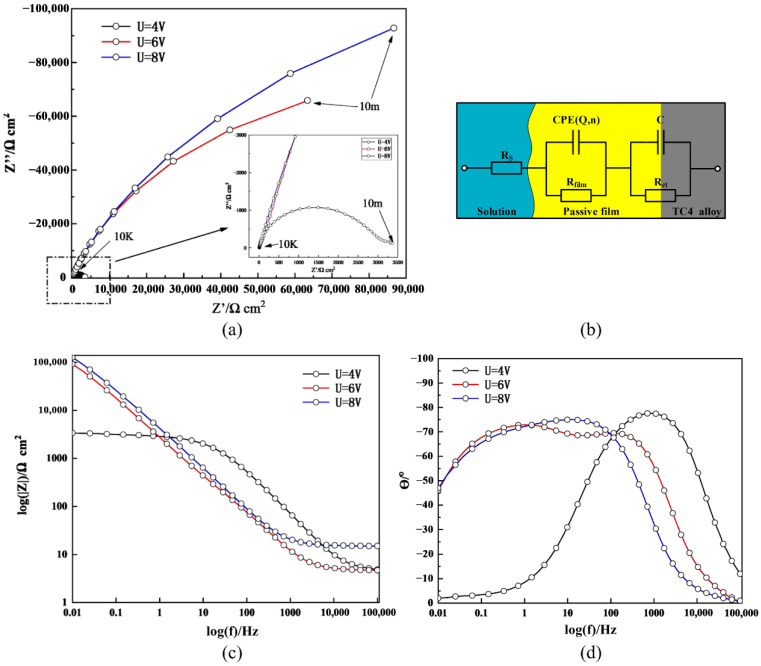
Electrochemical impedance spectroscopy (EIS) plots for TC4 in 10% NaNO_3_ solution. (**a**) Nyquist plots; (**b**) electrical equivalent circuit (EEC) used to fit the EIS data; (**c**) Bode impedance plots; (**d**) Bode phase plots.

**Figure 6 materials-15-08148-f006:**
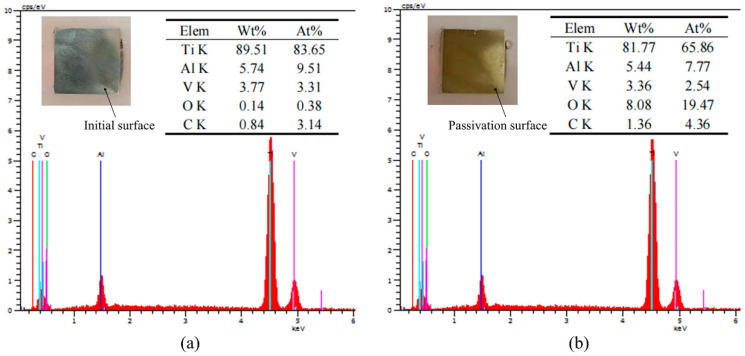
Energy-dispersive X-ray spectrometry (EDX) data for (**a**) initial surface and (**b**) passivated surface at 6 V for 2 min.

**Figure 7 materials-15-08148-f007:**
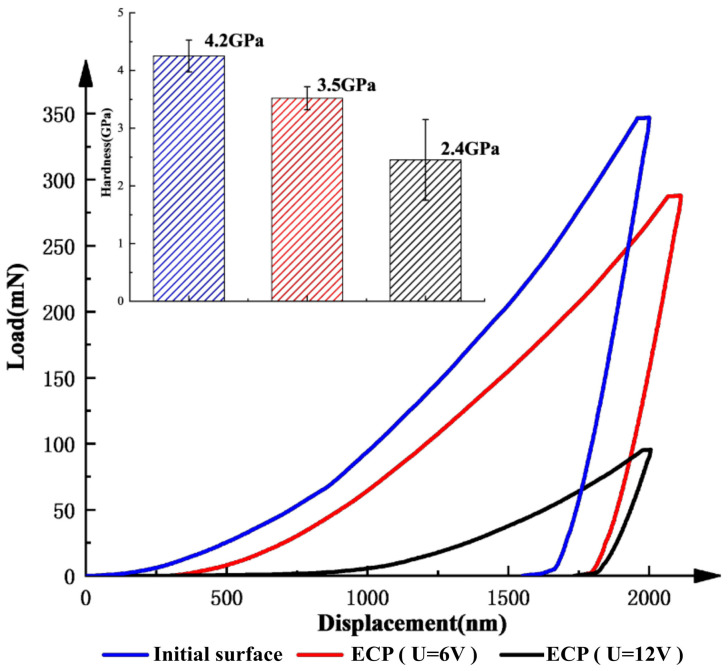
Measured load–displacement curves and calculated hardness values.

**Figure 8 materials-15-08148-f008:**
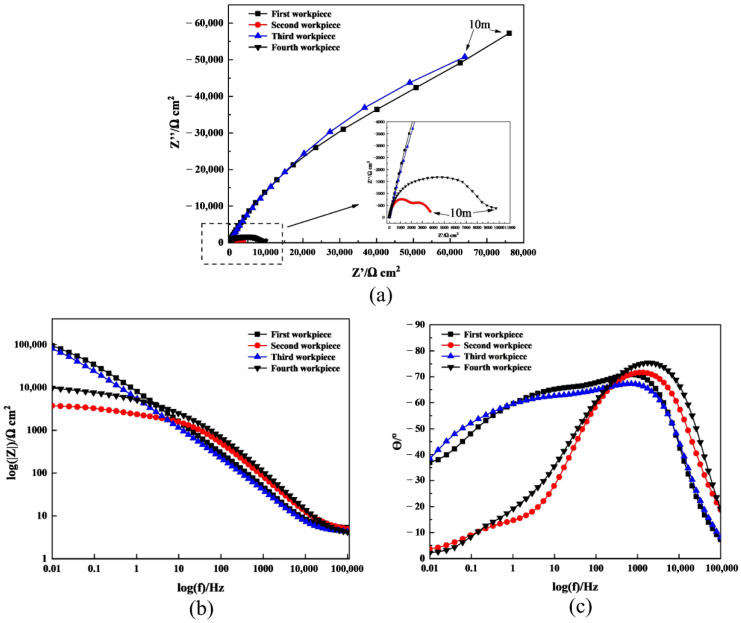
EIS curves for different pretreatment modes. (**a**) Nyquist plots; (**b**) Bode impedance plots; (**c**) Bode phase plots.

**Figure 9 materials-15-08148-f009:**
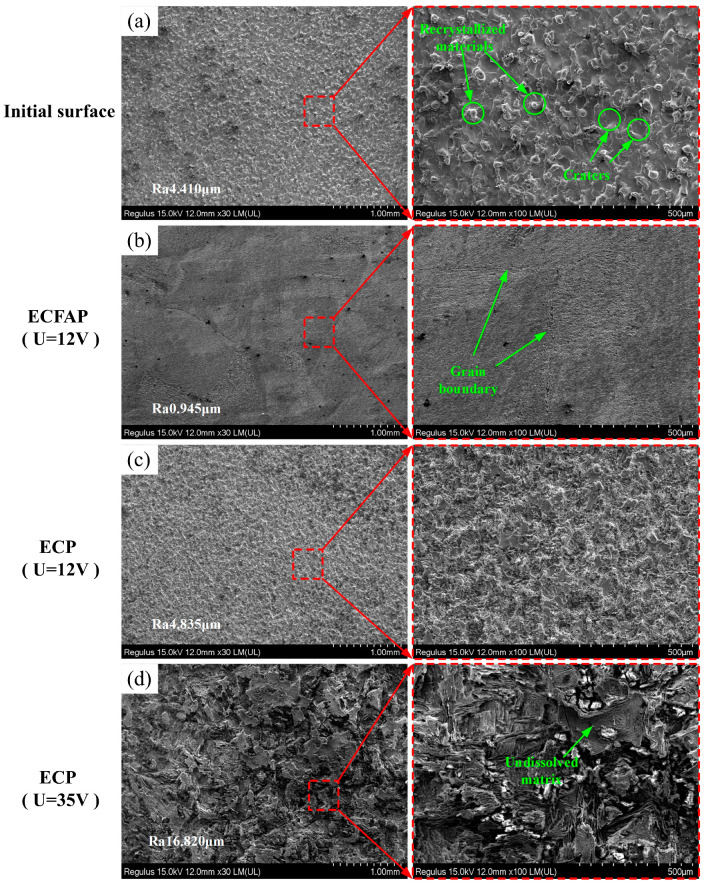
Comparison of surface finishing quality with and without treatment with abrasive particles. (**a**) Initial surface; (**b**) ECFAP (U = 12 V); (**c**) ECP (U = 12 V); (**d**) ECP (U = 35 V).

**Figure 10 materials-15-08148-f010:**
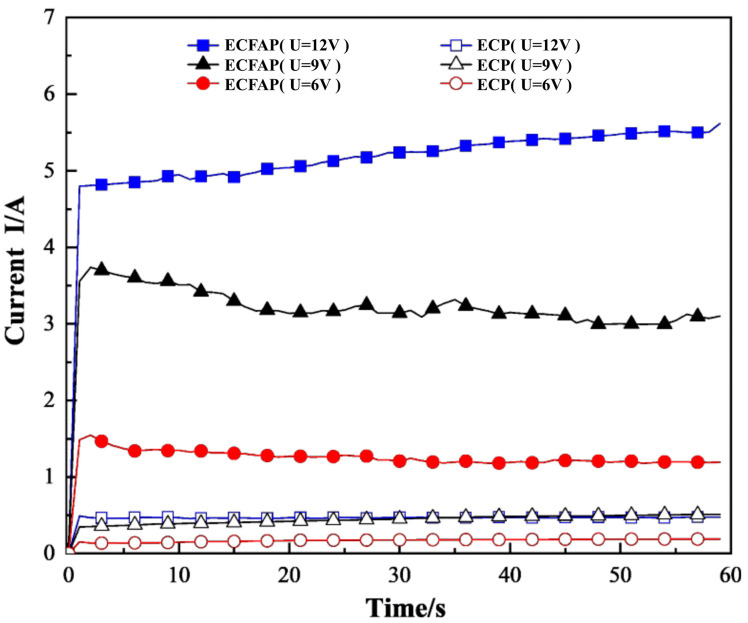
Comparison of current values measured during ECM and ECFAP treatments.

**Figure 11 materials-15-08148-f011:**
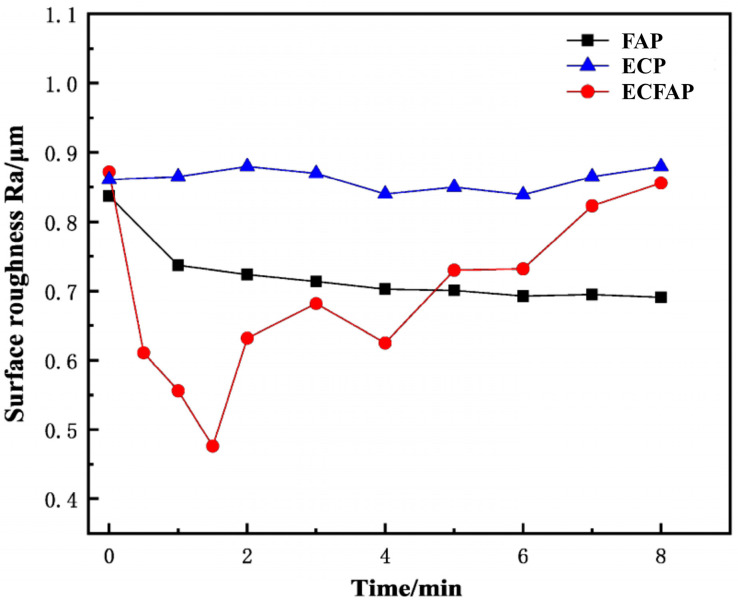
Time evolution of surface roughness of samples subjected to different polishing treatments.

**Figure 12 materials-15-08148-f012:**
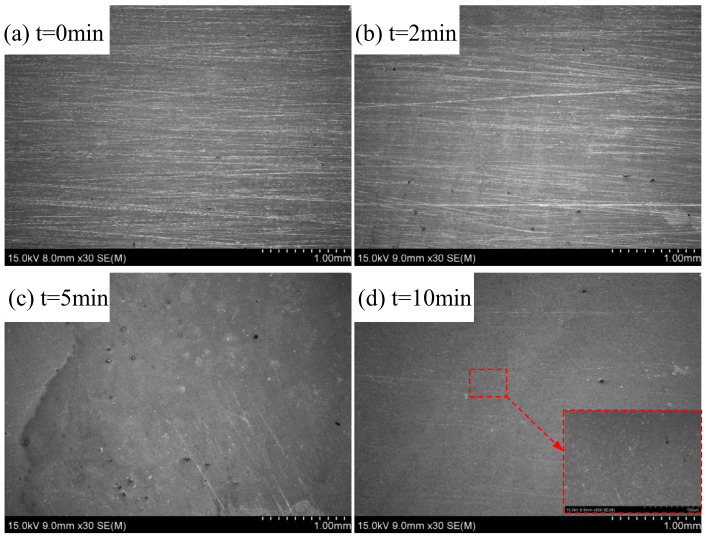
Material removal mechanism of ECFAP in region I (U = 3 V). (**a**) t = 0 min; (**b**) t = 2 min; (**c**) t = 5 min; (**d**) t = 10 min.

**Figure 13 materials-15-08148-f013:**
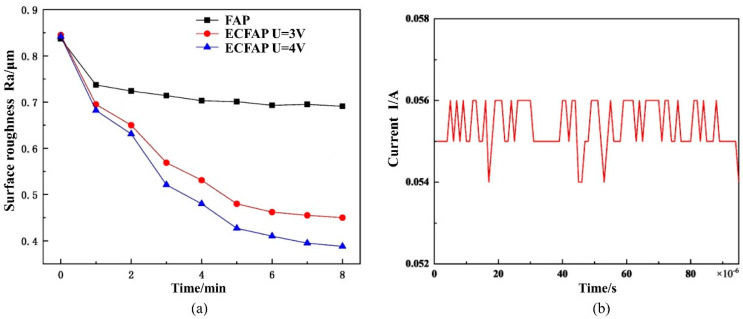
(**a**) Variation of surface roughness with time; (**b**) current vs. time curve during ECFAP (U = 3 V).

**Figure 14 materials-15-08148-f014:**
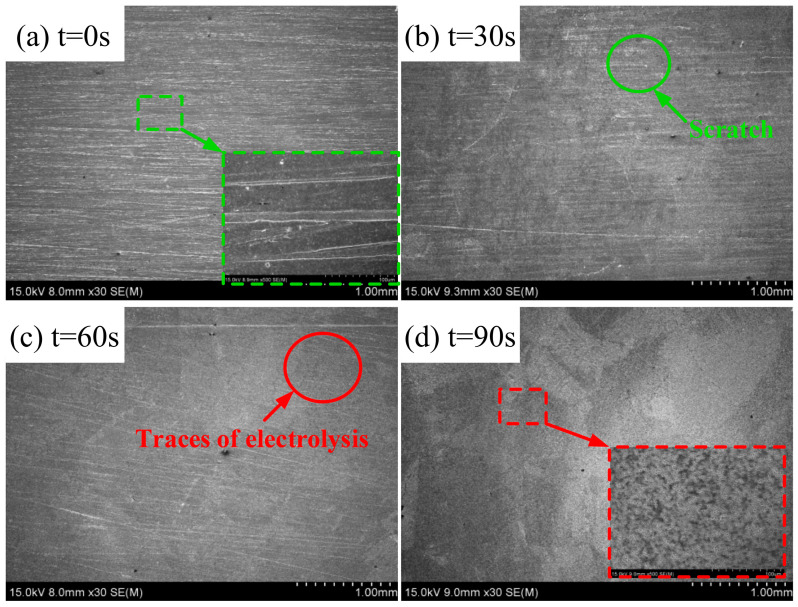
Material removal mechanism of ECFAP in region II (U = 6 V). (**a**) t = 0 s; (**b**) t = 30 s; (**c**) t = 60 s; (**d**) t = 90 s.

**Figure 15 materials-15-08148-f015:**
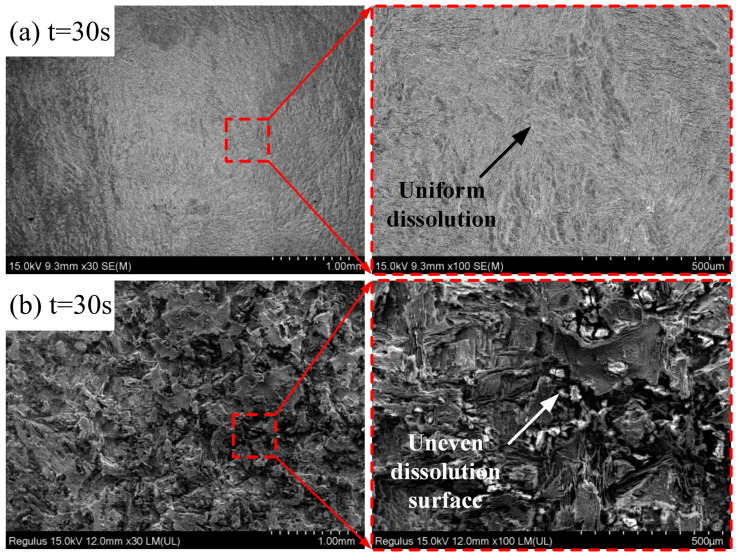
(**a**) ECFAP with U = 35 V for 30 s; (**b**) ECP with U = 35 V for 30 s.

**Figure 16 materials-15-08148-f016:**
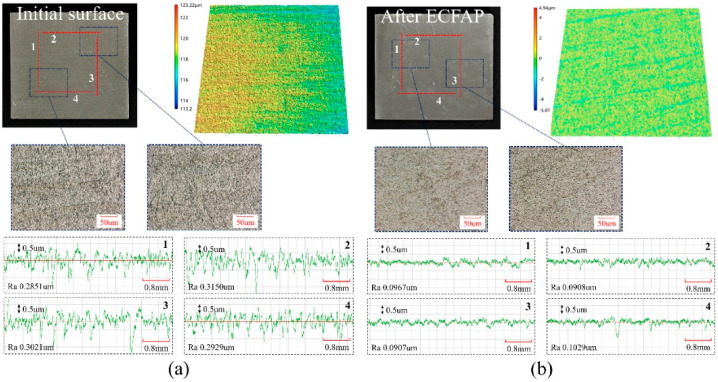
Comparison of samples before and after ECFAP: (**a**) 3D morphology and roughness profile of initial; (**b**) ECFAP-treated surfaces.

**Figure 17 materials-15-08148-f017:**
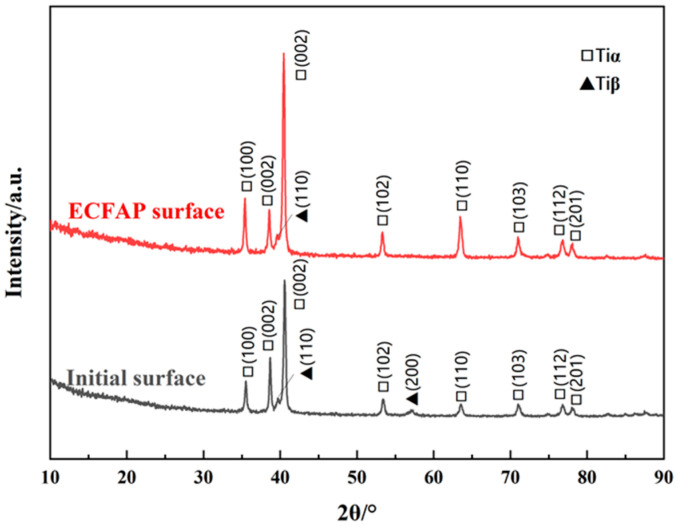
XRD patterns of the initial surface and after ECFAP treatment.

**Figure 18 materials-15-08148-f018:**
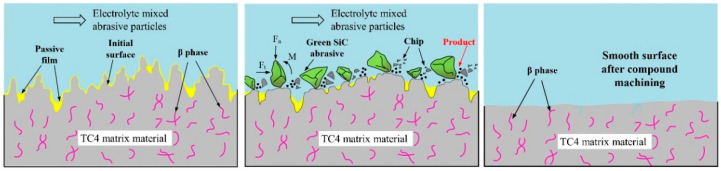
Model of material removal by ECFAP in 10% NaNO_3_.

**Figure 19 materials-15-08148-f019:**
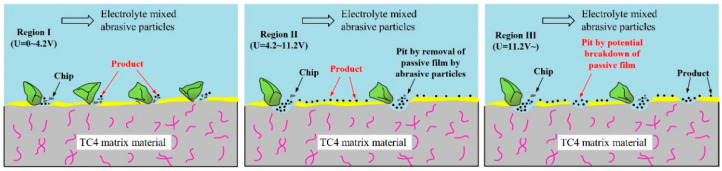
Models of material removal by ECFAP in different voltage ranges.

**Table 1 materials-15-08148-t001:** Comparison of different polishing technologies.

Processing Technologies	Structural Adaptability	Environmental Friendliness	Equipment Complexity	Polishing Efficiency
MB	Low	High	Simple	Low
AFM	High	High	Complex	High
CP	High	Low	Simple	Low
MAF	High	High	Complex	Low
LP	Low	High	Complex	Low
ECP	High	Low	Simple	High

**Table 2 materials-15-08148-t002:** Chemical composition of TC4 alloy [wt.%].

Alloy	Ti	Al	V	Fe	C
TC4	Balance	5.5–6.8	3.5–4.5	0.3	≤0.3

**Table 3 materials-15-08148-t003:** Experimental parameters of ECFAP tests.

Parameter	Value
Electrolyte	10 wt.% NaNO_3_ solution
Abrasive particles	10% (*v*/*v*) 500-mesh SiC abrasive particles
Machining gap	2 mm
Initial pressure	0.8 MPa
Flow speed	9.7 m/s
Voltage	3, 4, 6, 9, 12, 35 V

**Table 4 materials-15-08148-t004:** Fitting parameters of EEC model for TC4 in 10% NaNO_3_ solution.

Parameter	*R_s_* [Ω cm^2^]	*R_film_* [Ω cm^2^]	*Q* [μF/cm^2^]	*n*	*R_ct_* [Ω cm^2^]	*C* [μF/cm^2^]	*χ* ^2^
U = 4 V	4.624	2378	16.374	0.8469	752.2	5.026	0.003680
U = 6 V	4.638	173,800	79.381	0.8298	40.48	103.82	0.000509
U = 8 V	14.91	231,190	57.808	0.8427	1343	180.32	0.001282

**Table 5 materials-15-08148-t005:** Pretreatment methods used in EIS measurements.

Workpiece No.	Pretreatment Mode
1	ECP (5 s)
2	ECP (5 s) → FAP (5 s)
3	ECP (5 s) → FAP (5 s) → ECP (5 s)
4	ECP (5 s) → FAP (5 s) → ECP (5 s) → FAP (5 s)

**Table 6 materials-15-08148-t006:** EEC model fitting parameters of TC4 samples subjected to different pretreatments.

	*R_s_* [Ω cm^2^]	*R_film_* [Ω cm^2^]	*Q* [μF/cm^2^]	*n*	*R_ct_* [Ω cm^2^]	*C* [μF/cm^2^]	*χ* ^2^
1	4.533	113,890	32.458	0.77232	518.3	20.362	0.015096
2	4.201	1983	8.147	0.8399	1487	271.1	0.004976
3	3.601	116,820	48.886	0.74406	216.4	26.829	0.009161
4	3.385	3074	5.314	0.8654	4894	48.61	0.018525

## Data Availability

Relevant data are available from the corresponding author.

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
