# Peer review of "Electrochemical Polishing of Ti6Al4V Alloy Assisted by High-Speed Flow of Micro-Abrasive Particles in NaNO3 Electrolyte"

_materials, 2022, doi:10.3390/ma15228148_

Round 1

Reviewer 1 Report

Electrochemical polishing (ECP) is key in some applications, copper OFE, titanium, etc.

You missed some Works breakthrough working with bacteria helping the process, a idea that worked in copper alloys and in other , as it was demonstrated in https://doi.org/10.1016/j.jclepro.2014.01.061 and in 2-3 more, with comparison with other processes for finishing like burnishing (did you mention it?). Burnishing was and is key in many Surface applications, and in Titanium, after A.Rodriguez Works.

Your figures are OK, but you must discuss the processing time,  A key aspect in many industrial processes.

Why ref 1 if it is about laser? Much related are biomachining or burnishing alternatives.

Ti6Al4V: which one would be the most closer application in real industries?

Author Response

Reviewer’s comment 1

You missed some Works breakthrough working with bacteria helping the process, a idea that worked in copper alloys and in other, as it was demonstrated in https://doi.org/10.1016/j.jclepro.2014.01.061 and in 2-3 more, with comparison with other processes for finishing like burnishing (did you mention it?). Burnishing was and is key in many Surface applications, and in Titanium, after A.Rodriguez Works.

Response to comment 1

Thanks to the reviewer's reminder. We accept the reviewer's comments, add the relevant description in the introduction and add the literature citation. On the other hand, the polishing technologies has been summarized in a table for better understanding by the reader.

1) Location: chapter 1 (Introduction), page 1, line 31 - 32.

Traditional polishing technology mainly involves mechanical burnishing (MB) and grinding [7,8].

2) Location: chapter 1 (Introduction), page 2, line 45.

Table 1. Comparison of different polishing technologies.

Processing

 technologies

Structural

adaptability

Environmental

friendliness

Equipment

complexity

Polishing

efficiency

MB

Low

High

Simple

Low

AFM

High

High

Complex

High

CP

High

Low

Simple

Low

MAF

High

High

Complex

Low

LP

Low

High

Complex

Low

ECP

High

Low

Simple

High

3) Location: References, page 18, line 490 - 493.

  1. Egea, A. S.; Rodriguez, A.; Celentano, D.; Calleja, A.; de Lacalle, L. L. Joining metrics enhancement when combining FSW and ball-burnishing in a 2050 aluminium alloy. Surf. Coat. Technol. 2019, 367, 327-335.
  2. Aviles, A.; Aviles, R.; Albizuri, J.; Pallares-Santasmartas, L.; Rodriguez, A. Effect of shot-peening and low-plasticity burnishing on the high-cycle fatigue strength of DIN 34CrNiMo6 alloy steel. Int. J. Fatigue. 2019, 119, 338-354.

Reviewer’s comment 2

Your figures are OK, but you must discuss the processing time, A key aspect in many industrial processes.

Response to comment 2

Thanks for the reviewer’s comment. We add the description of time in the chapter of discussion, as follows:

1) Location: chapter 5 (Discussion), page 16, line 426 - 430.

The surface evolution with polishing time under different voltages has been discussed above. In short, at a small voltage, as the polishing time increases, the surface tends to be smooth and gradually tends to be stable. As shown in Figure 16, a better surface quality is obtained at a voltage of 3V and a processing time of 10 min.

2) Location: chapter 5 (Discussion), page 16, line 435 - 436.

With the extension of polishing time, the polishing effect decreased rapidly and then increased rapidly, indicating that short-time mixing polishing effect is better.

Reviewer’s comment 3

Why ref 1 if it is about laser? Much related are biomachining or burnishing alternatives. Ti6Al4V: which one would be the most closer application in real industries?

Response to comment 3

Thanks for the reviewer’s comment. In order to prevent readers from misunderstanding, the literature was replaced. On the current polishing technology, mechanical grinding and polishing in the practical industrial application of the most widely, this is beyond doubt. Especially with the emergence of industrial robot system, mechanical polishing has entered the climax of development. Of course, non-traditional polishing technology is also in continuous development. In the future, non-traditional polishing technology may have advantages for the processing of some special parts.

In the introduction, we rearranged the appearance order of polishing technologies, as follows:

1) Location: chapter 1 (Introduction), page 1, line 31 - 32.

Traditional polishing technology mainly involves mechanical burnishing (MB) and grinding [7,8].

2) Location: chapter 1 (Introduction), page 1, line 34 - 39.

To improve the efficiency and quality of the polishing process, many nontraditional and composite polishing technologies have been proposed, such as abrasive flow machining (AFM) [9–13], chemical polishing (CP) [14], magnetic abrasive finishing (MAF) [15–17], laser polishing (LP) [18,19], and electrochemical polishing (ECP) [20,21]. Table 1 lists the characteristics and applicable environments of these polishing technologies.

3) Location: References, page 18

  1. Zhu, W., Beaucamp, A. Compliant grinding and polishing: A review. Int. J. Mach. Tools Manuf. 2020, 158, 103634.
  2. Kumar, R.; Singh, S.; Aggarwal, V.; Singh, S.; Pimenov, D. Y.; Giasin, K.; Nadolny, K. Hand and Abrasive Flow Polished Tungsten Carbide Die: Optimization of Surface Roughness, Polishing Time and Comparative Analysis in Wire Drawing. Materials, 2022, 15, 1287.
  3. Rokosz, K.; Solecki, G.; Mori, G.; Fluch, R.; Kapp, M.; Lahtinen, J. Effect of polishing on electrochemical behavior and passive layer composition of different stainless steels. Materials 2020,13, 3402.

Reviewer 2 Report

Jia et al. have presented the manuscript titled: Electrochemical polishing of Ti6Al4V alloy assisted by high speed flow of micro-abrasive particles in NaNO3 electrolyte. Overall presentation of the work is good, but there are few suggestions which I think are necessary to explain before publication.

1.      Abstract is weak, I suggest the authors to add the final sentence about the practical implementation of this research work.

2.      In the introduction section, problem statement is missing, what was the purpose to carry out this research work? What was lacking in previous study, which has compelled the authors to perform this research work?

3.      In the whole manuscript I request the authors to revise the subscripts and superscripts in chemical formulas and units.

4.      In the literature review I suggest the authors to add the comparison study in the form of table with the previous performed research works.

5.      Why authors have just used the 10 wt.% NaNO3 solution, have the checked the other (5, 15) wt.% NaNO3 solution of electrolyte.

6.      In Figure 4b, please use the term of current density J, rather than i on Y-axis. Same for the Figure 4a.

7.      I suggest the authors to provide the XRD analysis of the sample.

Author Response

Reviewer’s comment 1

Abstract is weak, I suggest the authors to add the final sentence about the practical implementation of this research work.

Response to comment 1

Thanks for the reviewer’s comment. We accepted the reviewers' comments and added the following phrase to the abstract section of the revised manuscript:

Location: Abstract, page 1, line 21 - 23.

Finally, ECFAP-treated specimens with optimum voltage of 3 V for 10 min exhibited an average surface roughness of 0.0953 µm.

Reviewer’s comment 2

In the introduction section, problem statement is missing, what was the purpose to carry out this research work? What was lacking in previous study, which has compelled the authors to perform this research work?

Response to comment 2

Thanks for the reviewer’s comment. We accept the reviewers ' comments and add a description to the introduction. The details are as follows:

Location: chapter 1 (Introduction), page 2, line 78 - 80.

With the widespread use of additive manufacturing of complex structural parts, the demand for polishing in environments with poor tool accessibility is increasing. A convenient, efficient and low-cost polishing technology is urgently needed.

Reviewer’s comment 3

In the whole manuscript I request the authors to revise the subscripts and superscripts in chemical formulas and units.

Response to comment 2

Thanks for the reviewer’s comment. In the revised draft, we carefully reviewed and revised these format issues raised by the reviewers.

Reviewer’s comment 4

In the literature review I suggest the authors to add the comparison study in the form of table with the previous performed research works.

Response to comment 4

Thanks for the reviewer’s comment. We accept the reviewer’s comments and add tables to the introduction to compare different polishing methods.

Location: chapter 1 (Introduction), page 2, line 45.

Table 1. Comparison of different polishing methods.

Processing

 technologies

Structural

adaptability

Environmental

friendliness

Equipment

complexity

Polishing

efficiency

MB

Low

High

Simple

Low

AFM

High

High

Complex

High

CP

High

Low

Simple

Low

MAF

High

High

Complex

Low

LP

Low

High

Complex

Low

ECP

High

Low

Simple

High

Reviewer’s comment 5

Why authors have just used the 10 wt.% NaNO3 solution, have the checked the other (5, 15) wt.% NaNO3 solution of electrolyte.

Response to comment 5

Thanks for the reviewer’s comment. We make the following responses:

Through previous studies, 10 wt.% NaNO3 and 500-mesh abrasives were mixed in a certain proportion to obtain a better comprehensive polishing effect. Therefore, this paper focuses on the corresponding research work for 10 % solution and 500-mesh abrasive particles. The research on the matching between different solution concentrations and different abrasive particles will be carried out in subsequent research. Of course, we added a simple description in the manuscript. For example: According to the results of preliminary experiments, we selected 10 wt.% NaNO3 solution and 500-mesh SiC as the solution system

Reviewer’s comment 6

In Figure 4b, please use the term of current density J, rather than i on Y-axis. Same for the Figure 4a.

Response to comment 6

Thanks for the reviewer’s comment. We accept the reviewer’s comments and have revised Figure 4.

Location: chapter 4 (Results), page 6, line 199-201.

Figure 4. Polarization and cyclic voltammetry (CV) curves of TC4: (a) polarization curves under ECP and ECFAP conditions; (b) CV curve under ECP conditions.

Reviewer’s comment 7

I suggest the authors to provide the XRD analysis of the sample.

Response to comment 6

Thanks for the reviewer’s comment. We accepted the reviewer’s comments and added XRD analysis. A description of this part is added to the revised draft as follows:

Location: chapter 4 (Results), page 15, line 400-404; page 16, line 408-409.

After ECFAP treatment, the phase structure of the machined surface was detected by X-ray diffractometer (PANalytical X’pert, PANalytical, NL), as shown in Fig.1. It should be noted that there is little difference between ECFAP treatment and grinding surface. It can be inferred that there is no obvious residual electrolytic layer on the sample surface after ECFAP treatment.

Figure 17. XRD patterns of the initial surface and after ECFAP treatment.

Round 2

Reviewer 1 Report

Paper is Ok